

# Hyperoside alleviates doxorubicin-induced myocardial cells apoptosis by inhibiting the apoptosis signal-regulating kinase 1/p38 pathway

Lingxia Chen[1,2], Zhi Qin[1,2] and Zhong-bao Ruan[2]

[1] Department of Cardiology, Nanjing University of Chinese Medicine, Nanjing, China
[2] Department of Cardiology, The Affiliated Taizhou People's Hospital of Nanjing Medical University, Taizhou School of Clinical Medicine, Nanjing Medical University, Taizhou, China

## ABSTRACT

**Background**. Cardiotoxicity is a side effect of the anthracycline broad-spectrum anti-tumor agent, doxorubicin (DOX). Hyperoside, a flavonoid glycoside extracted from many herbs, has anti-apoptotic and anticancer properties. However, its impact on the alleviation of DOX-induced apoptosis in cardiomyocytes remains elusive.

**Methods**. The HL-1 cell line was treated with 100 μ M hyperoside for 1 h prior to treatment with 100 μ M hyperoside and 1 μ M DOX for 24 h. The cell counting kit-8 (CCK-8) assay was used to detect cell viability; DCFH-DA fluorescent probe was used to detect (reactive oxygen species) ROS; biochemical methods were used to detect the activity of glutathione (GSH), catalase (CAT), superoxide dismutase (SOD), malondialdehyde (MDA); the degree of apoptosis following DOX insult was assessed using immunofluorescence staining and terminal deoxynucleotidyl transferase mediated deoxy uridine triphosphate nick end labeling (TUNEL) assay; the change in protein expression of apoptosis signal-regulating kinase 1 (ASK1), p38, and apoptosis markers was determined using western blot.

**Results**. Hyperoside ameliorated DOX-induced oxidative stress in HL-1 cells, up-regulated GSH, SOD and CAT activity, reduced ROS production and inhibited MDA overproduction. Moreover, in addition to promoting HL-1 cell apoptosis, DOX administration also increased B-cell lymphoma (Bcl)-2-associated X-protein and cleaved caspase-3 protein levels and decreased Bcl-2 protein level. Hyperoside therapy, however, significantly reversed the impact of DOX on the cardiomyocytes. Mechanically, DOX treatment increased the phosphorylation of the ASK1/p38 axis whereas hyperoside treatment attenuated those changes. In a further step, hyperoside synergizes with DOX to kill MDA-MB-231 cells.

**Conclusions**. Hyperoside protects HL-1 cells from DOX-induced cardiotoxicity by inhibiting the ASK1/p38 signaling pathway. Meanwhile, hyperoside maintained the cytotoxicity of DOX in MDA-MB-231 cells.

Corresponding author
Zhong-bao Ruan, tzcardiac@163.com

## INTRODUCTION

Doxorubicin (DOX) is an anthracycline antibiotic used in clinical practice for tumor chemotherapy (*Herrmann, 2020*; *Sawicki et al., 2021*). Research has revealed that DOX has a strong binding affinity for topoisomerase and accumulates in cardiomyocytes as an isomer, causing severe DOX-induced cardiotoxicity (DIC) (*Sangweni et al., 2022*). Clinically, one-quarter of patients undergo chemotherapy-induced DIC in a dose-dependent manner, which severely restricts the clinical application of DOX (*Swain, Whaley & Ewer, 2003*). There are various mechanisms of DIC, among which oxidative stress, mitochondrial dysfunction, and apoptosis have been most widely reported (*Wallace, Sardão & Oliveira, 2020*; *Wenningmann et al., 2019*). Therefore, pharmacological interventions are required in these patients to attenuate DIC.

Apoptosis signal-regulated kinase 1 (ASK1) is a type of reactive oxygen species (ROS)-sensitive mitogen-activated protein kinase (MAP3K) that activates p38 and c-Jun N-terminal kinase (JNK) pathways *via* a cascade connection mediated by MAP kinase kinases to cause cell apoptosis. Evidence has reported that the ROS-ASK1-p38 pathway is significant for the treatment of cardiomyocyte apoptosis using DOX (*Jiang et al., 2020*). Additionally, *Gao et al. (2014)* reported that DOX-induced oxidative stress caused the phosphorylation of ASK1 and p38, podocyte damage, and ultimately, renal injury in murine podocytes. Inhibition of the ROS-ASK1-p38 pathway decreases apoptosis in conditions such as subarachnoid hemorrhage (*Gao et al., 2022*), non-alcoholic steatohepatitis (*Lan et al., 2022*), and diabetic cardiomyopathy (*Ding et al., 2021*). Hyperoside, a flavonoid glycoside extracted from various herbs, such as Hypericaceae and Rosaceae, has several cytoprotective properties, including anti-apoptosis, antioxidation, and anticancer property (*Wang et al., 2022a*; *Wang et al., 2022b*; *Ferenczyova, Kalocayova & Bartekova, 2020*). Hyperoside reduces mitochondrial dysfunction in podocytes during DOX-induced kidney injury (*Chen et al., 2017*). Additionally, hyperoside has been reported to reduce damage and oxidative stress in human keratinocyte cells by negatively regulating the p38 signaling pathway (*Charachit et al., 2022*). However, whether hyperoside can protect cardiomyocytes from DOX-induced apoptosis and its underlying mechanism have not yet been fully elucidated.

In our work, we explored the impact of hyperoside on DIC in an HL-1 cardiomyocyte cell line.

## MATERIALS & METHODS

### Materials

Hyperoside was purchased from MedChemExpress (Monmouth Junction, NJ, USA) as a crystalline powder with a purity of 99.56%. It was dissolved in DMSO and prepared into a mother liquor of 100 mM. DOX was purchased from MedChemExpress as a crystalline powder with a purity of 99.48%. It was dissolved in DMSO and prepared as a 10 mM mother liquor. They were diluted to the desired concentration when used. One-step terminal deoxynucleotidyl transferase deoxyuridine triphosphate nick end labeling (TUNEL) apoptosis assay kit, reactive oxygen species (ROS) assay kit and malondialdehyde (MDA) content assay kit purchased from Beyotime Biotech Inc (Shanghai, China). Catalase (CAT)
activity assay kit, superoxide dismutase (SOD) activity assay kit and reduced glutathione (GSH) content assay kit purchased from Solarbio Science & Technology Co., Ltd. (Beijing, China). Cell counting kit-8 (CCK-8) was purchased from Apexbio (Houston, TX, USA). Fetal bovine serum (Bovine serum) was purchased from Sigma (Santa Clara, CA, USA). HL-1 cells and MDA-MB-231 cells were obtained from Shanghai Fuheng Biotechnology Co., Ltd. (Shanghai, China).

## Cell culture and treatments

HL-1 cells were cultured in the Claycomb media (Sigma, St. Louis, MO, USA) containing 10% fetal bovine serum, 2 mM L-glutamine (Solarbio, Beijing, China), 100 U/mL penicillin (Gibco, Waltham, MA, USA) and 100 μg/mL streptomycin (Gibco, Waltham, MA, USA) at 37 °C. Cells were randomly allocated into four groups, *i.e.,* the control, hyperoside, DOX and DOX + hyperoside groups. Cells in the control group were cultured in the medium for 24 h; cells in the hyperoside group were cultured in a medium supplemented with 100 μM hyperoside for 24 h; cells in the DOX group were cultured in a medium supplemented with 1 μM DOX for 24 h; cells in the DOX + hyperoside group were pretreated with 100 μM hyperoside for 1 h and then treated with 100 μM hyperoside and 1 μM DOX together for 24 h.

## Test for cell viability

CCK-8 test was used to detect cell viability. To determine the optimal concentration of DOX that can lead to a survival rate close to but not exceeding, 50%, we used DOX of different concentrations (0 μM, 0.1 μM, 0.5 μM, 1 μM, 2 μM, 3 μM, 4 μM, 5 μM, and 10 μM) to treat the HL-1 cells. To determine the safe contentration range of hyperoside that will not cause severe cell toxicity, we treated HL-1 cells with hyperoside at concentrations of 0 −500μ M (0 μM, 5 μM, 10 μM, 50 μM, 100 μM, 200 μM, 300 μM, 400 μM and 500 μM). Further, hyperoside of final concentrations (0 μM, 25 μM, 50 μM, 100 μM, 125 μM, 150 μM, and 175 μM) was used to examine whether it will inhibit DOX-induced cell injuring in HL1 cells. After incubation for 24 h, ten microliters of CCK-8 reagent were added into each well of a 96-well plate and then incubated at 37 °C for two hours. The absorbance of optical density (OD) value at 450 nm was measured using a microplate reader (Bio Tek, Winooski, Vermont, USA). Each treatment was performed in triplicate. The percentage of cell viability for each group was calculated as follows: cell viability rate = OD of the experimental group/OD of the control group $\times 100$.

## Immunofluorescence (IF) staining

Cells were fixed with 4% paraformaldehyde (Solarbio, Beiijng, China) for 15 min and further blocked with an immunofluorescent blocking solution (Beyotime Biotech Inc, Jiangsu, China) for 1 h at 15–25 °C. Next, the cells were incubated with anti-cleaved caspase-3 (1:100, AF7022, Affinity, San Franscisco, CA, USA) at 4 °C overnight. And the anti-cleaved caspase-3 was diluted by antibody dilution (NCM Biotech Co., Ltd, Newport, RI, USA). Samples were incubated with goat anti-rabbit secondary antibody (immunoglobulin G) (diluted by PBS, 1:500, A23420, Abbine) for 1 h at room temperature

in the dark, followed by sealing with the Antifade Mounting Medium containing 4,6-diamidino-2-phenylindole (DAPI) (Solarbio, Beijing, China). Images were captured using a fluorescence microscope (Leica, Oskar, Germany), and fluorescence intensity was quantified using the ImageJ software.

### TUNEL staining

According to the manufacturer's protocol, a one-step TUNEL apoptosis assay kit (Beyotime, Jiangsu, China) was used to assess apoptosis in HL-1 cells and MDA-MB-231 cells. Cells were fixed with 4% paraformaldehyde for 30 min and further permeabilized in 0.3% Triton X-100 for 5 min at 15–25 °C. And then the cells were incubated with TUNEL reagent for 1 h at 37 °C in the dark. The samples were examined with a fluorescence microscope after being counterstained with DAPI. The fluorescence intensity was further analyzed using ImageJ software. The ratio of TUNEL-positive nuclei to total DAPI-stained nuclei was calculated to assess the apoptotic rate.

### Detection of ROS level

According to the manufacturer's protocol, a ROS assay kit was used to assess ROS level in HL-1 cells. In short, the cells were incubated with culture medium containing DCFH-DA fluorescent probe and hoechst 33,342 staining solution for live cells (Beyotime, Jiangsu, China) reagent for 30 min at 37 °C in the dark. The samples were examined with a fluorescence microscope. Then the fluorescence intensity was further analyzed using ImageJ software.

### Measurement of SOD, CAT, GSH and MDA levels

MDA content assay kit, CAT activity assay kit, SOD activity assay kit and GSH content assay were used to measure GSH, CAT, SOD and MDA levels. All operations are in accordance with the manufacturer's protocol. In short, after extracting GSH, MDA, CAT and SOD, respectively, the absorbance of OD value was measured using a microplate reader And the GSH, MDA, CAT and SOD levels were calculated according to the formulas provided in the instructions.

### Western blot (WB) analysis

Total protein was extracted from HL1 cells using radio immunoprecipitation assay lysis buffer (Beyotime, Jiangsu, China) containing phenyl methane sulfonyl fluoride (PMSF, MCE, USA), phosphatase inhibitor. Then an enhanced bicinchoninic acid protein assay kit (Beyotime, Jiangsu, China) was used to quantify the concentration of total protein. Equal amounts of protein were separated using sodium dodecyl sulfate-polyacrylamide gel electrophoresis (SDS-PAGE), transferred to polyvinylidene fluoride membranes (Millipore, Burlington, MA, USA), and blocked with NcmBlot blocking buffer (NCM Biotech Co., Ltd, Newport, RI, USA) at 15–25 °C for 10 min. The membranes were incubated with the primary antibody at 4 °C overnight and probed with the secondary antibody at room temperature for 1 h. Protein bands were observed under a gel documentation system (SYNGENE, Cambridge, UK). Quantitative analysis of the proteins was conducted using the ImageJ software. The primary and secondary antibodies used in WB analyses
were as follows: anti-t-ASK1 (1:1000, ab45178, Abcam, Cambridge, UK), anti-p-ASK1 (1:10000, ab278547, Abcam, Cambridge, UK), anti-t-p38 (1:2000, ab170099, Abcam, Cambridge, UK), anti-p-p38 (1:1000, ab195049, Abcam, Cambridge, UK), anti-B-cell lymphoma (Bcl)-2-associated X-protein (Bax) (1:5000, ab32503, Abcam, Cambridge, UK), anti-Bcl-2 (1:10000, ab182858, Abcam, Cambridge, UK), anti-cleaved caspase-3 (1:1000, 9664, Cell Signaling Technology, Danvers, MA, USA), anti-p53 (1:1000, 2524, Cell Signaling Technology, Danvers, MA, USA) and anti-β-actin (1:10000, AF7018, Affinity, San Fransisco, CA, USA) antibodies, and horseradish peroxidase goat-anti-rabbit secondary antibody (1:10,000, YFSA02, YiFeiXue Biotechnology, Beijing, China). All the primary antibodies used were diluted by antibody dilution (NCM Biotech Co., Ltd, Newport, RI, USA) and the secondary antibody was diluted by 1× tris-buffered saline tween (TBST) buffer (Solarbio, Beijing, China). β-actin levels were taken as the control.

## Quantitative reverse transcription-polymerase chain reaction (qRT-PCR)

Total RNA from HL-1 cells was extracted using the TRIzol Universal Reagent (TIANGEN, Beijing, China) following the manufacturer's instructions. The total RNA concentration of each sample was then measured using nanodrop 2000 (Thermo Fisher Scientific, Waltham, MA, USA). Next, the complementary DNA (cDNA) was synthesized by using the All-In-One 5X RT MasterMix (ABM, G490, USA). The procedure was conducted following the manufacturer's protocol. In short, the reaction was executed at 25 °C for 10 min and then 42 °C for 15 min, followed by 5 min at 85 °C. Quantitative analysis of RNA was accomplished using the Roche LightCycler 480 (Roche, Basel, Switzerland) with BlasTaq™ 2X qPCR MasterMix (G891; ABM, New York, NY, USA). The reaction conditions were as follows: The reaction was executed at 95 °C for 3 min, followed by 40 cycles of 15 s at 95 °C and 60 s at 60 °C. The relative quantification of the target genes was normalized by GAPDH levels and calculated using the $2^{-\Delta\Delta CT}$ method. The primer sequences were presented in Table 1.

## Statistical analysis

GraphPad Prism 8 was used to analyse the data from at least three independent trials, which were indicated as mean ± standard error of mean. Multiple group comparisons were evaluated using one-way analysis of variance, and comparisons between two groups were evaluated using Student's $t$-test. The significance of statistics was defined as $P < 0.05$. *** $P < 0.001$, ** $P < 0.01$, * $P < 0.05$.

# RESULTS

## Optimal concentration of DOX and hyperoside for HL-1 cells

Cell survival rate was used to ascertain the optimal concentration of DOX and hyperoside. The CCK-8 result revealed that HL-1 cell survival rate steadily reduced with increasing DOX concentrations. Specifically, the cell viability was 63.12% at the concentration of 1 µM DOX, which was significantly different compared to that when the concentration of DOX was 0 µM (Fig. 1A). Additionally, we used IF staining to examine the expression level

**Table 1 Primer sequences used in this research.**

| Gene | sequences (5′–3′) | Tm (°C) |
|---|---|---|
| Mus Bax | F: AGACAGGGGCCTTTTTGCTAC | 54.4 |
| | R: AATTCGCCGGAGACACTCG | 53.6 |
| Mus Bcl2 | F: GCTACCGTCGTGACTTCGC | 55.8 |
| | R: CCCCACCGAACTCAAAGAA | 56.3 |
| Mus cleaved caspase-3 | F: CTCGCTCTGGTACGGATGTG | 56.5 |
| | R: TCCCATAAATGACCCCTTCATCA | 53.7 |
| Mus GAPDH | F: AGGTCGGTGTGAACGGATTTG | 62.6 |
| | R: GGGGTCGTTGATGGCAACA | 62.6 |

of cleaved caspase-3 in HL-1 cells after they were treated with DOX of different doses. The results showed that increasing DOX concentrations gradually raised cleaved-caspase 3 level. Specifically, when the concentration of DOX was 1 μM, the average densitometric value of cleaved caspase-3 was 39.98%, which was significantly different compared to that when the concentration of DOX was 0 μM (Figs. 1D and 1E). Therefore, we selected 1 μM as the ideal concentration of DOX to induce HL-1 cell damage. Further, we treated HL-1 cells with hyperoside at concentrations of 0–500 μM, and the CCK-8 result revealed that hyperoside at concentrations equal to or higher than 200 μM could cause significant changes in the survival rates of these HL-1 cells (Fig. 1B). Thus, we choose hyperoside at concentrations of 0–175 μM for further optimization. The result revealed that hyperoside improved the survival rate of HL-1 cells within a specific range from 50 μM to 100 μM; however, hyperoside treatment did not significantly increase cell viability with the further increases in its concentrations (Fig. 1C). Thus, 100 μM was selected as the optimal concentration of hyperoside for subsequent experiments in this study.

## Hyperoside diminished DOX-induced oxidative stress damage in HL-1 cells

To explore the potential mechanism of cardiomyocyte protection associated with hyperoside, markers related to oxidative stress injury were examined. DCFH-DA fluorescent probe was used to measure expression level of ROS, and the results suggested that hyperoside significantly inhibited DOX-induced ROS production in HL-1 cardiomyocytes (Figs. 2A, 2B). Further, The results of biochemical assays showed that DOX remarkably decreased the activity of antioxidant enzymes (GSH, CAT, SOD) in HL-1 cells, which rebounded after hyperoside pretreatment (Figs. 2C, 2D and 2E); MDA was overproduced in the DOX group and diminished in the DOX + hyperoside group (Fig. 2F). The above outcomes revealed that hyperoside can reduce ROS production, increase the activity of antioxidant enzymes, and consequently attenuate DOX-induced oxidative stress damage in HL-1 cardiomyocytes.

## Hyperoside attenuated DOX-induced apoptosis in HL-1 cells

DOX-induced apoptosis was visualized by the TUNEL staining of HL-1 cells. The fluorescence microscopic images and data analysis revealed that the number of TUNEL-positive cells expressing the green tunneling magnetoresistance signal was markedly

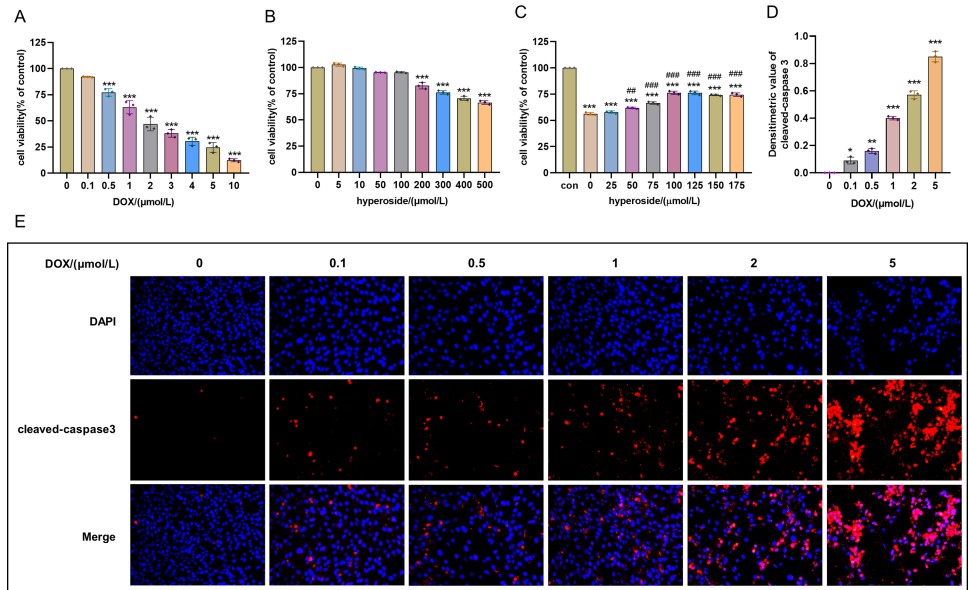

**Figure 1** **Optimal concentration of doxorubicin (DOX) and hyperoside for HL-1 cells.** (A) Effects of DOX on the survival rate of HL-1 cells. *** $P < 0.001$ compared with the 0-$\mu$M group. (B) Effects of hyperoside on the survival rate of HL-1 cells. *** $P < 0.001$ compared with the 0-$\mu$M group. (C) Effects of hyperoside on the survival rate of HL-1 cells induced by DOX. *** $P < 0.001$ compared with the control group; ### $P < 0.001$ compared with the 0-$\mu$M group. (D) Quantitative analysis of cleaved caspase-3 expression represented by bar graph ($n = 3$). *** $P < 0.001$, ** $P < 0.01$, * $P < 0.05$, compared with the 0-$\mu$M group. (E) Representative immunofluorescence labeling images for cleaved caspase-3 (red) in HL-1 cells. Data are indicated as mean ± standard error of the mean (SEM) from three independent experiments.

increased in the DOX group *versus* the control group. The results revealed a remarkable increase in apoptosis. Whether hyperoside treatment attenuated DOX-induced apoptosis was further explored. The results revealed that hyperoside treatment significantly reduced DOX-induced apoptosis in HL-1 cardiomyocytes, which was indicated by a remarkable decrease in TUNEL-positive cells (Figs. 3A and 3B). IF staining was used to detect cleaved caspase-3. The results revealed that cleaved caspase-3 expression was markedly increased following DOX treatment, and the effect was attenuated by hyperoside treatment (Figs. 3C and 3D).

## Hyperoside down-regulated apoptosis-related proteins in DOX-treated HL-1 cells

To investigate mechanisms of apoptosis following DOX treatment, qRT-PCR was used to identify Bax/Bcl-2 and cleaved caspase-3 mRNA levels of cells in each group. The results confirmed that hyperoside administration significantly reduced DOX-induced increased levels of Bax/Bcl-2 and cleaved caspase-3 (Figs. 4A and 4B). Further, we examined the levels of apoptosis-related proteins, and the WB results revealed significantly increased levels of Bax/Bcl-2 following DOX treatment whereas hyperoside treatment reversed the effect (Figs. 4C and 4D). Meanwhile, the administration of hyperoside significantly inhibited DOX-induced high expression of the cleaved caspase-3 protein (Figs. 4C and 4E). In

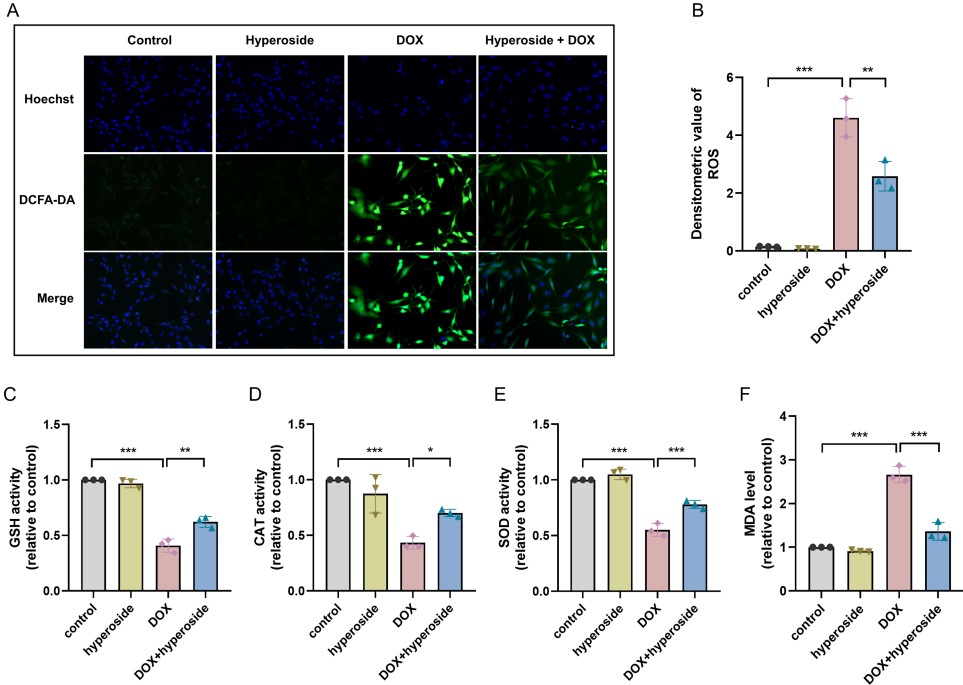

**Figure 2 Hyperoside attenuated doxorubicin (DOX)-induced oxidative stress injury in HL-1 cells.** (A) Representative fluorescence labeling images of DCFH-DA fluorescent probe to detect (reactive oxygen species) ROS (green) in HL-1 cells. Scale bars, 50 μm. (B) Quantitative analysis of ROS depicted by bar graph ($n = 3$). (C-F) Biochemical methods to detect glutathione (GSH), catalase (CAT), superoxide dismutase (SOD), malondialdehyde (MDA) levels in HL-1 cells. (C) Relative activity of GSH depicted by bar graph ($n = 3$). (D) Relative activity of CAT depicted by bar graph ($n = 3$). (E) Relative activity of SOD depicted by bar graph ($n = 3$). (F) Relative expression level of MDA depicted by bar graph ($n = 3$). *** $P < 0.001$, ** $P < 0.01$, * $P < 0.05$. Data are expressed as mean ± standard error of the mean (SEM) from three independent experiments.

addition, the level of p53 was markedly increased in DOX-treated HL-1 cardiomyocytes but falled back after hyperoside pretreatment (Figs. 4C and 4F).

## Hyperoside inhibited ASK1/p38 signaling pathway in DOX-treated HL-1 cells

To explore the possible signaling pathways involved in the above apoptosis pathway. We explored the ASK1/p38 signaling pathway and the role of hyperoside in regulating this pathway. According to WB analyses, the phosphorylation levels of ASK1 and p38 were considerably elevated following DOX treatment; however, the administration of hyperoside markedly reversed the effect of this treatment (Figs. 5A–5C).

## Hyperoside promotes the anticancer property of DOX

Human breast cancer MDA-MB-231 cell was used to detect the impact of hyperoside on DOX-treated cancer cells. As the CCK-8 result revealed that DOX can cause dose-dependent damage to MDA-MB-231 cells, the higher the concentration of DOX, the worse the cell viability (Fig. 6A). Of interest, hyperoside displayed a synergistic cytotoxic response to DOX when treated with MDA-MB-231 cells (Fig. 6B). Further, TUNEL staining results

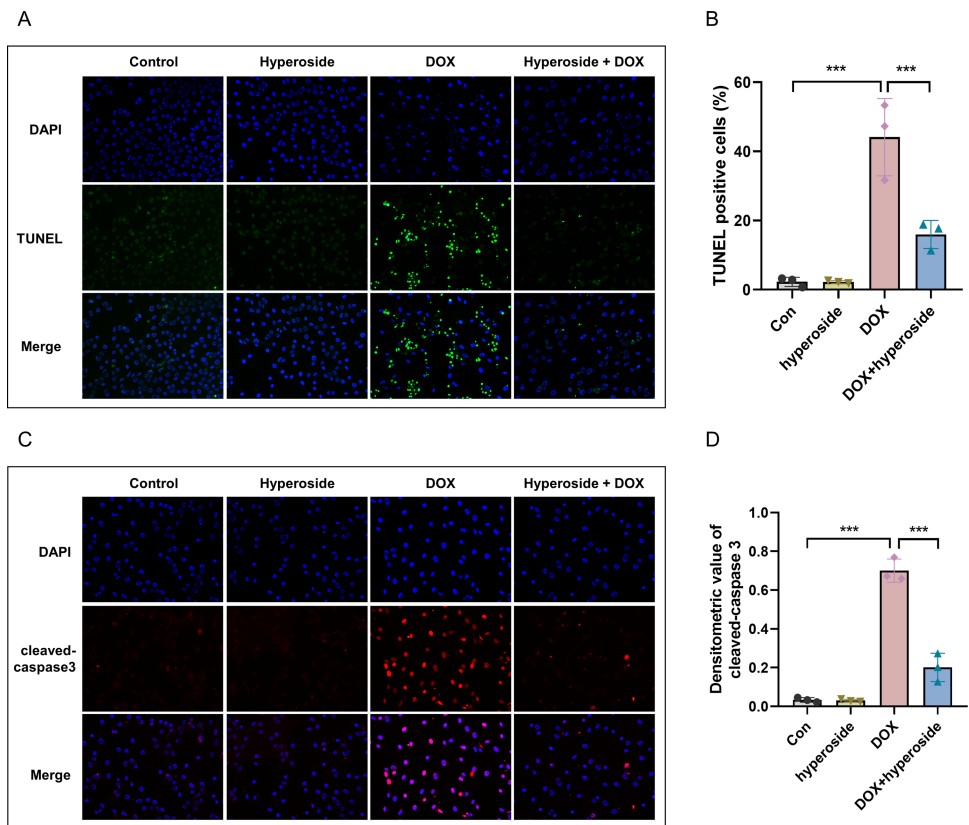

**Figure 3** **Hyperoside attenuated doxorubicin (DOX)-induced apoptosis in HL-1 cells.** (A) Representative fluorescence labeling images for terminal deoxynucleotidyl transferase deoxyuridine triphosphate nick end labeling (TUNEL) (green) in HL-1 cells. Scale bars, 50 μm. (B) Quantitative analysis of TUNEL-positive cells depicted by bar graph ($n = 3$). (C) Representative immunofluorescence labeling images of cleaved caspase-3 (red) in HL-1 cells. Scale bars, 50 μm. (D) Quantitative analysis of cleaved caspase-3 depicted by bar graph ($n = 3$). *** $P < 0.001$. Data are expressed as mean ± standard error of the mean (SEM) from three independent experiments.

showed that hyperoside accentuated DOX-induced apoptosis in MDA-MB-231 cells (Figs. 6C, 6D). The findings above suggested that hyperoside alleviated DIC while promoting the cytotoxicity of DOX on cancer cells.

## DISCUSSION

In our study, we identified the importance of hyperoside in inhibiting DOX-induced HL-1 cell apoptosis by restraining the activation of the ASK1/p38 signaling pathway. Doxorubicin is a commonly used clinical antitumor drug. However, due to its low selectivity of therapeutic targets, DOX can cause toxic side effects on liver (*Al-Qahtani et al., 2022*), kidney (*Wu et al., 2023*), nerve (*Orabi et al., 2021*), etc. in patients while treating tumors, the most serious of which is cardiotoxicity (*Chen, Shi & Dai, 2022*). DOX can cause acute, subacute, and chronic irreversible cardiotoxicity, eventually leading to heart failure (*Sheibani et al., 2022*). After stimulation with 1 μM DOX for 24 h, we found

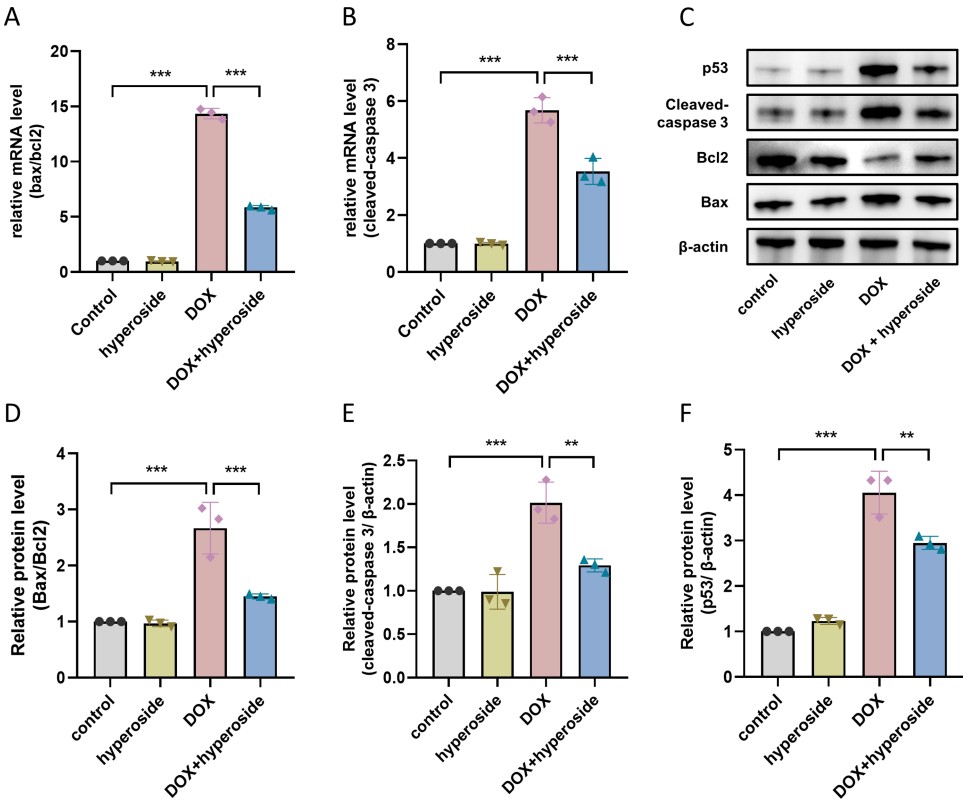

**Figure 4** **Hyperoside decreased the expression levels of apoptosis-related proteins in DOX-treated HL-1 cells.** (A) Relative mRNA level of Bcl-2 associated X-protein (Bax)/Bcl-2 in HL-1 cells ($n = 3$). (B) Relative mRNA level of cleaved caspase-3 in HL-1 cells ($n = 3$). (C) The expression of cleaved caspase-3, Bcl-2, Bax and p53 in HL-1 cells assessed by western blot analysis. $\beta$-actin was used as the control. (D) Quantitative analysis of Bax/Bcl-2 depicted by bar graph ($n = 3$). (E) Quantitative analysis of cleaved caspase-3 depicted by bar graph ($n = 3$). (F) Quantitative analysis of p53 depicted by bar graph ($n = 3$). *** $P < 0.001$, ** $P < 0.01$. Data are expressed as mean ± standard error of the mean (SEM) from three independent experiments.

decreased survival rate and increased apoptosis of HL-1 cells, indicating that DOX induced cardiotoxicity. *Zhang, Liu & Liu (2021)* reported that hyperoside alleviated LPS-induced cardiomyocyte death, and similarly in the present study, survival rate and apoptosis rate of HL-1 cells were decreased after hyperoside pretreatment, tentatively demonstrating its cardioprotective effect.

Former reports have revealed that DOX is converted to semi-quinone DOX (SQ-DOX) by the action of uncoupled nitric oxide synthase, NADPH oxidase and etc. SQ-DOX readily reduces oxygen molecules ($O_2$) to superoxide anions ($O_2$-), which are further transformed to hydrogen peroxide ($H_2O_2$) with the assistance of SOD (*Octavia et al., 2012*). $H_2O_2$ can generate highly reactive hydroxyl radicals (·OH) catalyzed by ferrous ions (*Kajarabille & Latunde-Dada, 2019*). Moreover, the aforementioned ROS will oxidize with DNA, RNA, proteins, and lipids in cells, damaging the subcellular structure and causing apoptosis and damage to cardiomyocytes. SOD, CAT and GSH are important

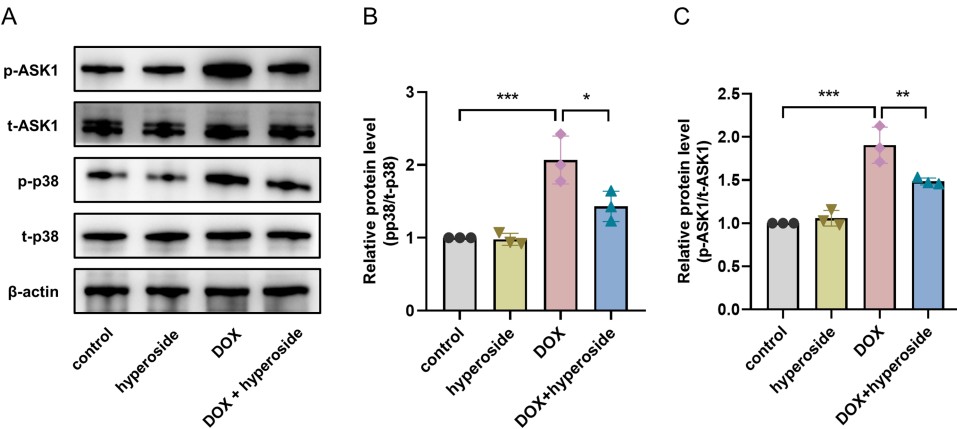

**Figure 5 Hyperoside inhibited apoptosis signal-regulating kinase 1 (ASK1)/p38 signaling pathway in doxorubicin (DOX)-treated HL-1 cells. $\beta$-actin was used as the control.** (A) The expression of p-ASK1, t-ASK1, p-p38, and t-p38 in HL-1 cells examined by western blotting. (B) Quantitative analysis of p-ASK1/t-ASK1 depicted by bar graph ($n = 3$). (C) Quantitative analysis of p-p38/t-p38 depicted by bar graph ($n = 3$). *** $P < 0.001$, ** $P < 0.01$, * $P < 0.05$. Data are expressed as mean $\pm$ standard error of the mean (SEM) from three independent experiments.

endogenous antioxidant enzymes that reflect the antioxidant ability of an organism (*Jing et al., 2020*; *Xiang et al., 2022*). MDA is the final product, also an indirect biomarker, of lipid peroxidation damage (*Zhang et al., 2011a*; *Zhang et al., 2011b*). Previous studies have reported that DOX-treated cardiomyocytes produce excessive ROS, accumulate MDA, and cause a decrease in endogenous cardiac antioxidant enzymes (*Wang et al., 2022a*; *Wang et al., 2022b*; *Pharoah et al., 2023*; *Sun et al., 2022*). Thus, excessive production of ROS and decayed activity of antioxidant enzymes are essential pathological characteristics of DIC. Similar to these findings, we detected a large production of ROS, a significant increase in MDA content, and a downregulation of SOD, GSH, and CAT in DOX-treated HL-1 cells, indicating that DOX caused oxidative stress injury in cardiomyocytes. Interestingly, the changes in ROS and MDA were attenuated after hyperoside pretreatment, while the levels of SOD, GSH, and CAT also rebounded. It is suggested that hyperoside enhanced the antioxidant capacity of cardiomyocytes, thus alleviating DOX-induced cardiotoxicity. *He et al. (2021)* found that hyperoside down-regulated MDA levels and increased SOD activity in an *in vivo* hypoxia model in mice. *Shah et al. (2013)* found that DOX treatment attenuated GSH and SOD activities and up-regulated lipid peroxidative value level in *in vivo* experiments of rats. And hyperoside markedly reversed these changes and enhanced the antioxidant capacity of rat hearts. These studies demonstrated that hyperoside also has significant antioxidant property *in vivo* in animals.

Myocardial apoptosis is a crucial mechanism of DIC. The Bcl-2 family plays a significant role in caspase activation and apoptosis (*Ma et al., 2020*; *Childs et al., 2002*). There are two categories of Bcl-2 family: anti-apoptotic proteins Bcl-2, Bcl-xL, etc. and pro-apoptotic proteins Bax, Noxa, etc (*Rosa et al., 2022*). As an important marker of apoptosis, caspase-3 is identified to be a significant regulatory molecule of the apoptosis

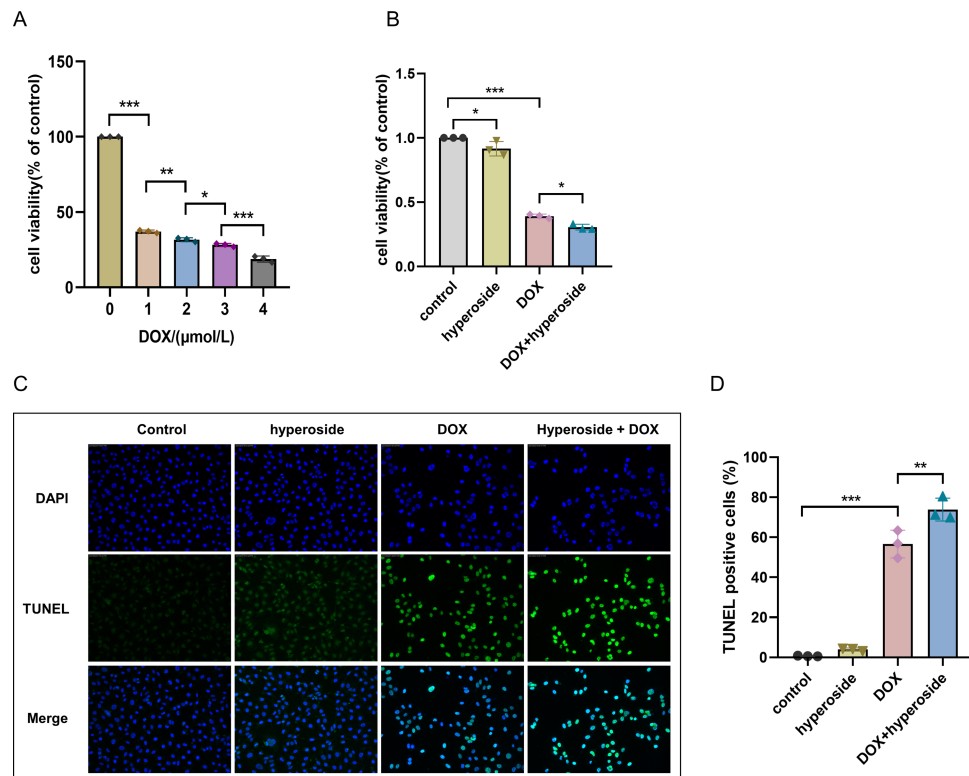

**Figure 6 Hyperoside promotes the anticancer property of DOX.** (A) CCK-8 assayed the effect of DOX on the survival rate of MDA-MB-231 cells. (B) CCK-8 assayed the effects of hyperoside on the survival rate of MDA-MB-231 cells induced by DOX. (C) Representative fluorescence labeling images for terminal deoxynucleotidyl transferase deoxyuridine triphosphate nick end labeling (TUNEL) (green) in MDA-MB-231 cells. Scale bars, 50 μm. (D) Quantitative analysis of TUNEL-positive cells depicted by bar graph ($n = 3$). *** $P < 0.001$, ** $P < 0.01$, * $P < 0.05$. Data are expressed as mean $\pm$ standard error of the mean (SEM) from three independent experiments.

pathway (*Li, Wang & Zhou, 2022*). Different stimulation signals initiate apoptosis *via* various pathways, however, the ultimate common effect is to activate caspase-3. DOX administration causes excessive oxidative stress and further activates the intrinsic apoptotic process, by upregulating Bax, caspase-3 and downregulating Bcl-2, eventually causing cardiomyocyte death (*Wenningmann et al., 2019*; *Li et al., 2021*). More specifically, our study discovered that DOX treatment can increase expression of cleaved caspase-3 and Bax, while decreasing expression of Bcl-2, and these changes can be reversed by hyperoside. The result suggests that hyperoside can reduce cardiomyocyte apoptosis, further suggesting a protective property of hyperoside on DIC. *Xiao et al. (2017)* found that hyperoside decreased the expression of Bax, cleaved caspase-3, and upregulated the level of Bcl-2 in a hypoxia/reoxygenation model of rat cardiomyocytes, demonstrating its anti-apoptotic property, which is in accordance with our findings. A previous research has reported that DOX can upregulate tumor suppressor gene p53 level, which can regulate Bcl-2 family proteins (*e.g.*, Noxa, Bax) (*Yu & Zhang, 2005*), change the Bax/Bcl-2 ratio, causing mitochondria to permeabilize and release cytochrome c and caspase, leading

to cardiomyocyte apoptosis (*Zhang et al., 2011a*; *Zhang et al., 2011b*; *Zhang et al., 2012*). Consistent with the above literatures, we determined that DOX treatment can increase expression of p53, and hyperoside can significantly reverse the change.

Mitogen activated protein kinase (MAPK) is considered to be a crucial target molecule that induces apoptosis (*Das et al., 2011*). The MAPK pathway has 32 main branching pathways, of which the p38 MAPK cascade is mainly engaged in the response of cells to stress (*Keshet & Seger, 2010*). Numerous studies have validated the close relationship between the p38 MAPK cascade and DIC. Activated p38 can induce cardiomyocyte apoptosis by activating the downstream molecules Bax, Bcl-2, and p53 (*Thandavarayan et al., 2010*; *Guo et al., 2013*; *Spallarossa et al., 2006*). ASK1, a key regulatory factor in the MAPK cascade pathway, is sensitive to oxidation and reduction. It can be activated under oxidative stress conditions and then triggers apoptosis *via* phosphorylation of JNK and P38 (*Jalmi & Sinha, 2015*; *Dai et al., 2020*). *Jiang et al. (2020)* have demonstrated that DOX can promote p38 phosphorylation levels by activating ASK1 signaling pathway, leading to cardiomyocyte apoptosis. Several reports have validated that the inhibition of the ASK1/p38 signaling pathway alleviated the pathological damage caused by apoptosis (*Li et al., 2022*; *Wu et al., 2018*). In this study, hyperoside dramatically reduced dox-induced phosphorylation of ASK1 and p38, in keeping with the above reports. The result suggested that hyperoside is a regulator engaged in the negative control of ASK1/p38 activation and that DOX-induced myocardial apoptosis may be related to ASK1/p38.

Hyperoside is a kind of flavonol glycosides with antitumor property. On lung cancer cells *in vitro*, hyperoside downregulated Bcl-2 level, upregulated Bax level, and increased the expression of anti-tumor factors such as p53. In addition, *in vivo* experiments in mice have also demonstrated its inhibitory effect on tumor angiogenesis (*Liu et al., 2016*). *Fu et al. (2016)* revealed that hyperoside exerted anti-cancer function by inducing autophagy in lung cancer cells. Further, this might be related to the inhibition of Akt/mTOR/p70S6K signaling pathway and the activation of ERK1/2 pathway. Also in breast cancer, hyperoside inhibited the activity and migratory capacity of breast cancer cells, and activated mitochondrial apoptosis pathway by inhibiting ROS/NF-κB signaling pathway, which in turn induced breast cancer cell apoptosis (*Qiu et al., 2019*). Furthermore, in paclitaxol-treated MDA-MB-231 cells, hyperoside inhibited cell activity and facilitated apoptosis. And it also enhanced the sensitivity of MDA-MB-231 cells to paclitaxel *via* blocking TLR4 signaling. Interestingly, meanwhile, hyperoside inhibited paclitaxel-induced cytotoxicity in human normal mammary epithelial cells MCF-10A (*Sun et al., 2020*). Our research confirmed that hyperoside synergized with DOX to inhibit survival rate and induce apoptosis of MDA-MB-231 cells, but the specific mechanism needs further research.

## CONCLUSIONS

Our study suggests that hyperoside may reduce DOX-induced cardiomyocyte damage *via* inhibition of the ASK1/p38 signaling pathway, while raising the sensitivity of cancer cells to DOX. These findings revealed the significant value for the attenuation of DOX toxicity in normal cells and the synergistic effect of hyperoside in tumor cells. Due to these properties,

hyperoside may be a potential candidate for alleviating cardiotoxicity during DOX therapy. Our study showed a protective role of hyperoside on myocardium and provided insight into how hyperoside regulates cardiomyocyte apoptosis of DIC. However, hyperoside needs to be comprehensively evaluated in DIC animal models.

## ACKNOWLEDGEMENTS

We are grateful for the technical support from the Central Lab of Taizhou People's Hospital, China. We would also like to thank TopEdit for its linguistic assistance during the preparation of this manuscript.

### Funding
The authors received no funding for this work.

### Competing Interests
The authors declare there are no competing interests.

### Author Contributions
- Lingxia Chen conceived and designed the experiments, performed the experiments, analyzed the data, prepared figures and/or tables, authored or reviewed drafts of the article, and approved the final draft.
- Zhi Qin performed the experiments, prepared figures and/or tables, and approved the final draft.
- Zhong-bao Ruan conceived and designed the experiments, authored or reviewed drafts of the article, and approved the final draft.

### Data Availability
The data is available at figshare:

Chan, Lara (2023): Raw Data (New). figshare. Dataset. https://doi.org/10.6084/m9.figshare.22337665.v1

Chan, Lara (2023): Blots (New). figshare. Dataset. https://doi.org/10.6084/m9.figshare.22337656.v1

Chan, Lara (2023): supplement. figshare. Dataset. https://doi.org/10.6084/m9.figshare.22337695.v1

### Supplemental Information
Supplemental information for this article can be found online at http://dx.doi.org/10.7717/peerj.15315#supplemental-information.

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
