# Peer review of "Hyperoside alleviates doxorubicin-induced myocardial cells apoptosis by inhibiting the apoptosis signal-regulating kinase 1/p38 pathway"

_PeerJ, doi:10.7717/peerj.15315_

## Round 0.1 · original submission · Major Revisions

The reviewers have provided major comments.

·

Basic reporting

The work is clear and well contextualized. The authors have worked well to be concise and clear in their work and the language is very professional.

Some additional discussion of the chemistry involved in how hyperoside functions (are there others than anti-oxidative effects and where does it scale in that effect relatively?) or acknowledgement of understudied areas may be beneficial to the readers.

More detail should be provided on reagent purity and specific diluents used in a study of this nature.

Notations of percent should be out of 100 in Figure 1.

Please include controls in Figure 4 A.

Experimental design

Claycomb media is the ideal media to be used with HL-1 cells to have them most closely emulate heart tissue. It is advisable that a pilot be done to ensure the results remain consistent to confirm results in this media. If it is in short supply, it is possible to generate one's own from the recipe in "Hl-1 Cardiomyocytes as a model system" from March 2004.

It is not clear why the pretreatment with 100 hyperoside was used prior to Dox treatment, please elaborate.

Treatment with hyperoside alone would also be important to demonstrate in your model to show clear indications of tolerance.

Validity of the findings

Findings seem well supported and clearly stated.

Reviewer 2 ·

Basic reporting

1)The language is clear and unambiguous.

2) Line 209: "hyperoside has been reported to have cardioprotective properties". This information needs a citation.
Other than that literature references are well provided.

3) Raw data are well presented.

4) All the studies performed with different concentrations of agents, should be given in the materials and methods section instead of results (For ex: Lİnes 153-156). In the results section, only mentioning the selected concentrations and how they were selected would be enough.

Experimental design

1) Regarding the IF and WB:
-Please indicate the specie of the serum was raised (Goat serum) in the text.
-Please indicate the catalog numbers of the primary and secondary antibodies used in IF.
-Please indicate the diluent solution for the antibodies and if you used a commercial diluent, please mention the manufacturer.
-Please menion that you used ß-actin as the control in WB.

2) Regarding the qPCR:
-Please mention the RNA quantification method.
-Please indicate the cDNA synthesis protocol.
-Please mention how many cycles you had run.
-Please indicate that GAPDH was used as the housekeeping gene.
-Please mention how you calculated the quantific results (delta ct, fold change calculations).
-Instead of writing them plainly, please create a table for PCR primers and also indicate the Tm(℃) values for each primer.

3) Please also indicate the p value ranges for *, **, and *** in the statistics section.

Validity of the findings

1) Authors had mentioned oxidative stress a couple of times as an important factor. Why you did not performed total oxidant and anti-oxidant capacity assays? If possible, adding these assays would add strength to your article and support your hyphothesis. If not, please mention it as a limitation or discuss it with the findings of other similar studies which had performed TOC, TAC analyses.

2) There are several studies for hyperoside's effects on some cancer types which could be discussed in this article as well (For ex: https://www.nature.com/articles/aps2015148, https://www.mdpi.com/1422-0067/21/1/131, https://www.sciencedirect.com/science/article/abs/pii/S0753332216305534 ). The findings of these articles may be aded to the discussion as well.

3) There are a number of studies regarding the protective effects of hyperoside on cardiomyocytes for different conditions and addressing different pathways (For ex.: https://www.spandidos-publications.com/10.3892/mmr.2021.11925, https://www.sciencedirect.com/science/article/abs/pii/S0378111917306017, https://www.sciencedirect.com/science/article/pii/S0753332221003097). Discussing the outcomes of these (or any other) research and indicating similarities or dissimilarities would add stregth to your discussion. What other pathways were thought to play a role for its rpotective effect? How protective it was for different cases? Are there any in vivo experiments demonstrating this effect as well?

4) There's a study using "Hypericum hircinum" on doxorubicin-induced cardiotoxicity in rats (https://www.tandfonline.com/doi/abs/10.1080/14786419.2012.724409). The Hypericum hircinum contains hyperoside, along with some other pharmacologically important compounds. I believe authors should examine this research througoutly and discuss its outcomes by comparing their research.

Additional comments

Authors had investigated the effects of hyperoside-DOX combination on cardiomyocytes to indicate hyperoside's protective effects on cardiomyocytes on DIC. The study reveals that hyperoside reduces apoptosis by inhibiting the activation of the ASK1/p38 signaling pathway. Although the experimental design is solid and the results are well provided, there is still room for improvements in the discussion section. Also, some additions should be made in the materials and methods section as well.

Best regards.

Annotated reviews are not available for download in order to protect the identity of reviewers who chose to remain anonymous.

·

Basic reporting

Chen and colleagues present a manuscript about hyperoside alleviates the side effect of doxorubicin treatment to the cardiotoxicity. They used HL-1 cell as the model cell line to test the protection of DOX treatment. This is an issue of great importance to clinical research and patient treatments. However, there are several important questions that need to be addressed to demonstrate the reliability of their study.
1. Authors should test whether DOX treatment at the concentrations used in this manuscript are useful for cancer cells (at least one tumor type).
2. Authors should test whether hyperoside also protects tumor cells during the DOX treatment.
3. Bax was used as the apoptosis marker in this manuscript. However, BAX, NOXA, PUMA are downstream genes or p53 apoptotic pathway. Authors should also detect p53 pathway rather than only p38.

Experimental design

This is original primary research.
Research question well defined, relevant and meaningful. However, authors should do some deeper research to fix the concerns listed in the Basic reporting.

Validity of the findings

Conclusions are well stated but need further verification to better explain why the combination are necessary.

Additional comments

no

---

## Round 0.2 · accepted · Accept

One reviewer and I both agree to accept this revised paper. And the other two reviewers suggested minor comments, which have been well-revised by the authors. Taken together, this revised version could be accepted.

Reviewer 2 ·

Basic reporting

Thank you for your effort on revising the manuscipt for the used language. There are only a few typos which could easily be corrected while proof reading.

Experimental design

Used primary antibodies and their dilutions are valid. Authors should include the catalog numbers of the used kits for ROS, CAT, MDA, etc assays as well.

Validity of the findings

Used metholodolgy is improved and adequate after the antioxidant assays.

Additional comments

No comments.